# Network motifs shape distinct functioning of Earth's moisture recycling hubs

Nico Wunderling ®[1,2,3,5] ✉, Frederik Wolf[1,5], Obbe A. Tuinenburg[4] & Arie Staal ®[4] ✉

Earth's hydrological cycle critically depends on the atmospheric moisture flows connecting evaporation to precipitation. Here we convert a decade of reanalysis-based moisture simulations into a high-resolution global directed network of spatial moisture provisions. We reveal global and local network structures that offer a new view of the global hydrological cycle. We identify four terrestrial moisture recycling hubs: the Amazon Basin, the Congo Rainforest, South Asia and the Indonesian Archipelago. Network motifs reveal contrasting functioning of these regions, where the Amazon strongly relies on directed connections (feed-forward loops) for moisture redistribution and the other hubs on reciprocal moisture connections (zero loops and neighboring loops). We conclude that Earth's moisture recycling hubs are characterized by specific topologies shaping heterogeneous effects of land-use changes and climatic warming on precipitation patterns.

Life on land relies on the precipitation flows that provide a steady source of freshwater for most areas on Earth. This source enables a resilient Earth system and a safe operating space for humanity[1]. The origins of precipitation over land are almost equally distributed between Earth's land areas and oceans[2-4]. This implies that not only atmospheric changes such as climate warming, but also modifications at the land surface can affect precipitation patterns on Earth: land-cover transformations, including deforestation alter evapotranspiration flows, which may subsequently propagate across continents through evaporation-precipitation cycles[5-7]. Considering how critical they are for the Earth System, we have a surprisingly shallow understanding of how atmospheric moisture flows are arranged.

Recent improvements in atmospheric reanalysis data[8] and methodological advances to track the flows of moisture around the planet at high spatial and temporal resolutions[9] now allow for a significantly more detailed picture of these flows. Previously, we developed a dataset of monthly mean atmospheric moisture flows between each pair of 0.5° (ca. 55 km at the equator) grid cells across the globe based on all evaporation (including transpiration) on Earth during 2008–2017[3] (see Methods). One promising approach

to analyze these planet-encompassing data is as a spatial network in which each grid cell represents a node that may be connected either uni- or bidirectionally to any other node[10-13]. A directed link represents a mass flow from source to target, that is, a flow of moisture from its location of evaporation to that of precipitation. Such a network representation can reveal several local and global features of the underlying complex (climate) system[14-16]. Specifically, so-called *motifs* are local network structures that control how transitions may cascade across the network, and that enable assessing the sensitivity of the network to changes at certain nodes (i.e., grid cells) or edges (i.e., their links)[17-19]. Motifs are significantly over-expressed in real-world networks relative to random networks[17], enhancing information transport in many different types of complex networks such as the world-wide-web, gene expression networks, or food webs[18,20,21]. The enhanced information transport processes by the motifs can be positive in some contexts (in this case, for example, directed moisture transport allowing the forest to grow further downwind, which otherwise may not[6]), but also negative (deforestation causing loss of atmospheric moisture transport to downwind forests). Therefore, motifs in the moisture

[1]Earth System Analysis and Complexity Science, Potsdam Institute for Climate Impact, Research (PIK), Member of the Leibniz Association, 14473 Potsdam, Germany. [2]Stockholm Resilience Centre, Stockholm University, Stockholm SE-10691, Sweden. [3]High Meadows Environmental Institute, Princeton University, Princeton, NJ 08544, USA. [4]Copernicus Institute of Sustainable Development, Utrecht University, Utrecht 3584 CB, The Netherlands. [5]These authors contributed equally: Nico Wunderling, Frederik Wolf. ✉e-mail: nico.wunderling@pik-potsdam.de; a.staal@uu.nl

recycling network[5,12,19] may also provide novel insights into the structure and functioning of the global hydrological cycle.

By using network motifs, we find in this work that the Amazon Basin is a one-of-a-kind moisture recycling hub as compared to the other three main terrestrial moisture recycling hubs (the Congo Rainforest, South Asia, and the Indonesian Archipelago). Through the unique motif structure in the Amazon Basin, ecosystem damages caused by droughts, climate change, or deforestation may propagate particularly efficiently there.

## Results

### Hubs in the global moisture recycling network

Here we apply a network approach to high-resolution global atmospheric moisture flows. We build a directed, unweighted network[10,22,23] featuring the global atmospheric simulations[9]. Although $\rho = 25\%$ is a choice we made based on the moisture flow strength distribution (see Fig. S1) to focus on the regions of strongest moisture flows, our results are robust to substantial changes of $\rho$ (see sensitivity analyses in the Methods). By making this choice, we find the main moisture recycling hubs, which are relevant on an Earth system scale, and are of at least a sub-continental scale (similar to what has been used to define climate tipping elements[24]). As an initial step, we study the in-degree and out-degree of the moisture recycling network[25] (Fig. 1a, c), indicating the number of incoming and outgoing connections of each node. As expected, the tropics stand out as the region with the most intense atmospheric moisture transport regarding both in- and out-degree, reflecting the regions of highest precipitation and evaporation[3,26,27]. The large-scale tropical moisture recycling patterns are dominated by the Hadley Cell dynamics, with a small convergence zone with intense precipitation. The precipitation that falls in this convergence zone evaporated earlier in a relatively large zone in a band around the convergence zone[28]. Due to the large area that contributes evaporation to a small area of intense precipitation, the pattern of out-degree

differs systematically from that of in-degree, without marked regions that have outstanding out-degree maxima (Fig. 1a, b). Indeed, the stronger cutoff in the out-degree distributions (Figs. S2, S3) demonstrates a first significant finding: in-degree is consistently broader distributed than out-degree, indicating the existence of super-receivers but the absence of super-distributors of moisture. We study the dependency of regions on recycling-based moisture influx by considering particular link classes. Of special interest are moisture flows from land to land (Fig. 1c, d), as these not only serve as the main water supply to many regions but are also directly modulated by human activities that alter the land-surface moisture fluxes, such as large-scale deforestation and irrigation[27,29,30]. Comparing this with ERA5 precipitation and evaporation data on land, we mainly find the same patterns (see Fig. S4), which are broadly consistent with annual precipitation and evaporation values over land[8,26]. Globally, we identify four main hubs of large in- and out-degree in the land-to-land network: the Amazon Basin (AB, here defined in a broad sense, for example, including the Orinoco basin and the Guiana Shield), Congo Rainforest (CR), Indonesian Archipelago (IA) and South Asia (SA), see Fig. 2a. A deeper analysis of the topology of the network reveals the fundamentally differing functioning of these four regions.

### Motifs shape ecosystem functioning

We identify feed-forward loops (FFLs), zero loops (ZLs), and neighboring loops (NBrs; see explanatory insets in Fig. 2 and cf. ref. 19). FFLs involve three nodes linked in a directed triangular way, creating a lensing effect of moisture flow. This lensing effect is caused by a start node that is linked to a target node directly, but also via a one-node-detour through an intermediary node. Through this two-way interaction, the moisture flows from source to target are larger than they would appear if only direct flows were analyzed. ZLs consist of two nodes that are connected bidirectionally in a reciprocal manner. In other words, they show positive hydrological feedback between two

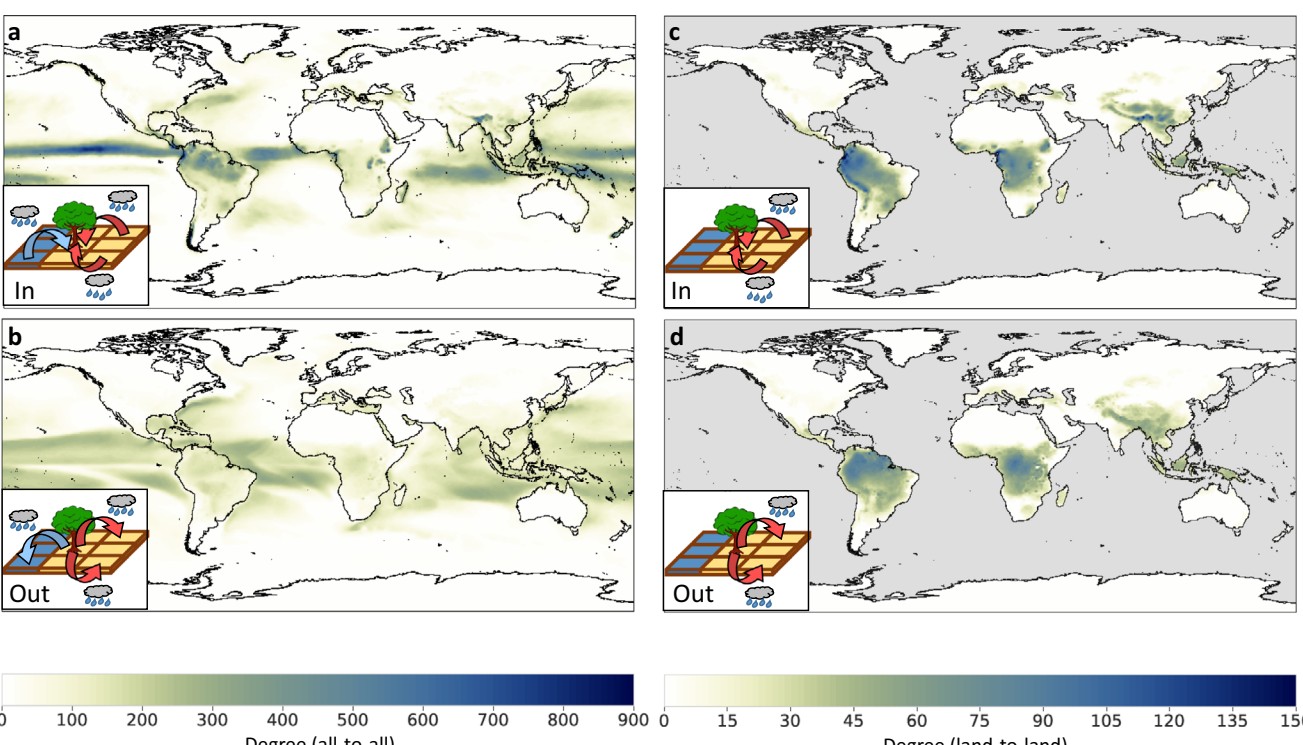

**Fig. 1 | In- and out-degree of the global moisture recycling network. a, b** In- and out-degree for all connections, and **c, d** solely for land-to-land interactions. High in-degree refers to nodes/grid cells receiving moisture from many other grid cells.

Elevated out-degree refers to moisture distributors. Insets illustrate the depicted measures.

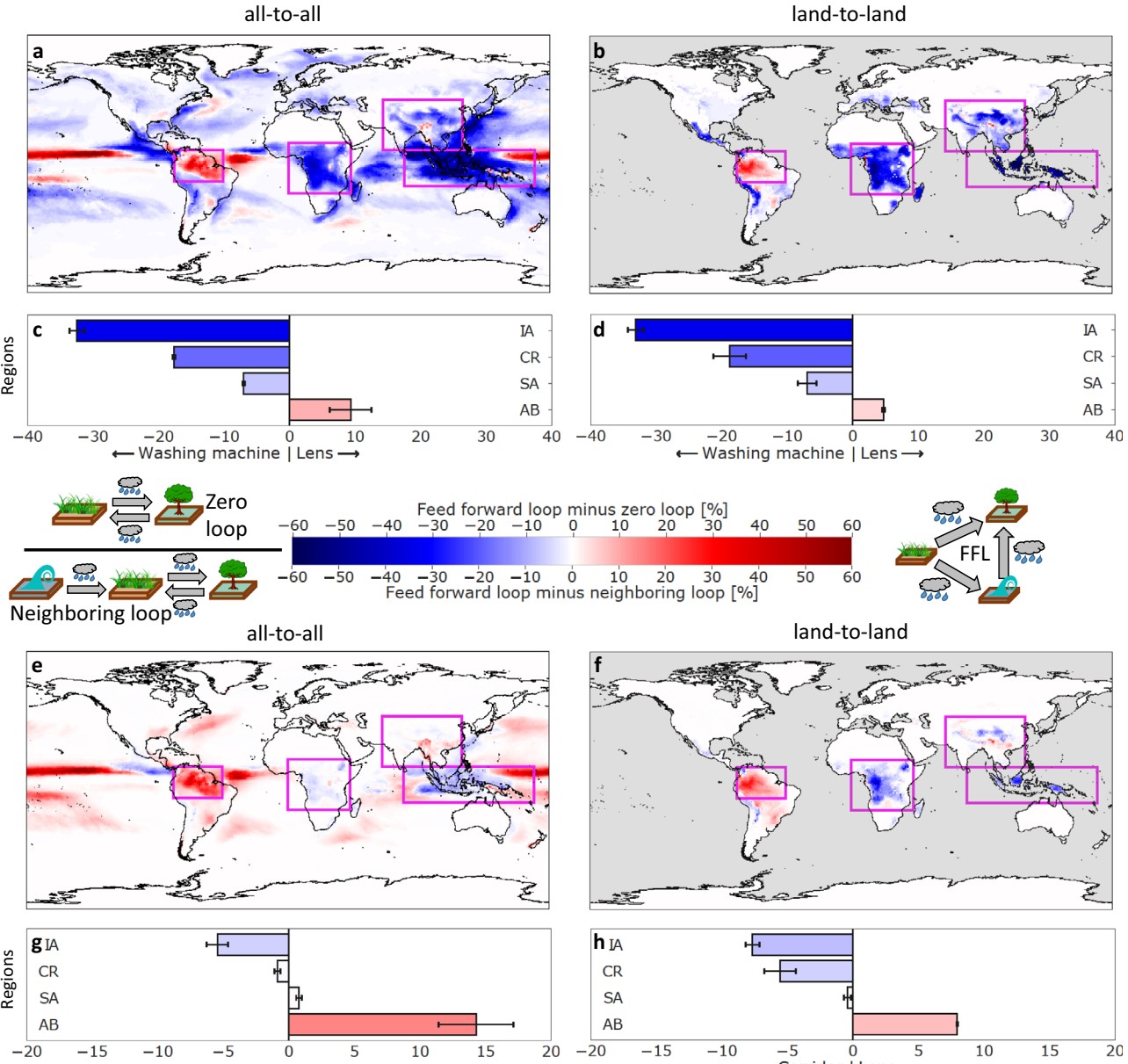

**Fig. 2 | Characterization of the Earth's moisture recycling hubs. a** FFL (feed-forward loop) and ZL (zero loop) strength difference for the all-to-all network and **b** for the land-to-land network. The corresponding spatially aggregated averages are shown in panels **c**, **d**. Illustrations next to the color bar schematically illustrate the motifs. The four main moisture recycling regions can be clearly distinguished. Redder colors represent larger lensing of moisture flows; bluer colors represent reciprocity of the network structure. Therefore, the Amazon basin (AB) is a directed lens, while the Indonesian Archipelago (IA), the Congo rainforest (CR), and South Asia (SA) are washing machines in descending order. **e**–**h** Same as **a**–**d** but for FFL

and NBr (Neighboring loop). Redder colors represent larger lensing effects and bluer colors indicate directed corridors (AB is a directed lens and IA is a directed corridor). The differences between the all-to-all network and the land-to-land network (mainly regarding AB and CR) highlight the differences between the ocean-to-land connectivity (which is included in the all-to-all network) and the organization of moisture recycling over land. The error bars are derived from a sensitivity analysis in which we analyze the moisture recycling network for a variable network threshold $\rho$ in the range from 20 to 30% (see SI Figs. S8–S11).

nodes in the network. NBrs indicate directed reciprocity: they describe a zero loop connected to an additional source node that induces directed moisture flow toward the ZL. Like FFLs, they involve three nodes and have a flavor of directedness in their topology. Thus, they share features of both node-to-node dependency and a lensing effect. NBrs imply that land cover or atmospheric change in the source node could disrupt the hydrological feedback between two other bidirectionally coupled nodes. In line with their functionality, we refer to spatial conglomerates of FFLs as *directed lenses*, to those of NBr as *directed corridors*, and to regions with an overwhelming fraction of ZLs as *washing machines* due to the circular nature of moisture flow.

Perturbations in directed lenses or corridors may propagate in a spatially predictable (directed) manner, whereas in the case of washing machines, they may spread in an unpredictable or perhaps spatially uniform way.

We find that edges and nodes are highly involved and locally organized in motifs: in the all-to-all network (land-to-land network), each edge is, on average, part of ~100 (~40) motifs, and each node is involved in ~5000 (~250) motifs. This implies that the network as a whole is highly locally-organized by motifs, and indeed, motifs appear approximately two orders of magnitude more often than we would expect from random networks (see SI and Figs. S14, S15, tested for a

threshold of $\rho = 25\%$). Going beyond classical degree measures, we compare the presence of FFLs and ZLs, and that of FFLs and NBrs (Fig. 2), both across the globe and for land flows only. Specifically, we study the per-node motif strength as relative values rescaled to their respective global maximum per-node count (for details, see Methods).

ZLs are ubiquitous across large parts of the planet (Fig. 2a–d), signifying widespread reciprocity in moisture flows. An oceanic exception can be found in the Pacific, where we observe how the Hadley Cell is represented in the network: rising air over the equator, falling air over the subtropics, and a surface return flow towards the equator result in many ZLs over the subtropics but dominating FFLs over the surface return flow areas (Fig. 2a).

Upon replacing ZLs with NBrs in the all-to-all network (Fig. 2e, cf. Fig. 2a), the FFLs become more important. This is a direct consequence of a large normalized number of ZLs in the overall network that are not connected to a third node. However, selecting the land-to-land connections only (Fig. 2f) leads to a considerable shift in the dominant network motif, as much fewer FFLs are present. This underlines that much of the land-to-land directedness in the global moisture recycling network is carried by NBrs and, thus, via moisture corridors. In contrast, much of the ocean-to-land connectivity is organized through FFLs via moisture lensing.

### Motif strength characterizes network hubs

When we compare the spatially aggregated average motif strength difference per focus region, an interesting difference emerges (Fig. 2c, d, g, h). From both the all-to-all and the land-to-land network, the same pattern arises: the Amazon is a *directed lens*, whereas especially in the Indonesian Archipelago and the Congo rainforest, the less directed motifs dominate. For the Amazon, it is known that effects of forest loss are propagated downwind, mainly westwards from the Atlantic Ocean towards the Andes and southwards out of the basin[31–34]. Our results reveal that the strong directedness of moisture propagation in the Amazon is globally one-of-a-kind, pointing at the unique role of Earth's largest rainforest in its regional hydrological cycle, and possibly its particular vulnerability[35]. However, the cause of this exceptional functioning is unclear. Potentially, it reflects the "biotic pump hypothesis," which states that the large-scale condensation of water vapor creates horizontal pressure differences in the lower atmosphere that propel local atmospheric dynamics[36–38]. Regardless of the mechanism, whether due to monthly or even very short-term diurnal effects[39], locations at the receiving end of an FFL will be particularly sensitive to deforestation at the source of the motif[30,31].

The other major tropical rainforests are characterized by reciprocal effects, indicated mainly by the ubiquity of ZLs. This suggests that these regions may exhibit a more local and diverse response to land-use changes or shifts in moisture distribution. Positive as well as negative changes in a certain location would affect that location not only directly but also indirectly. This may knock ecosystems more easily out of equilibrium, as loss of evaporation will cause a local loss in precipitation as well. However, it could also support the regrowth and reforestation of degraded or disturbed forests, as the resulting recovery of evaporation would recover precipitation locally as well. Put otherwise, the "hidden" local forest-precipitation feedback due to ZLs expands the local-scale hysteresis of tropical tree cover against precipitation levels that is believed to be widespread across the tropics[40].

Outside the tropics, South Asia is a hotspot of ZLs. This may be related to the Himalayas, given that other mountain regions across the globe have relatively many ZLs as well (see Fig. 2 globally and Fig. S16 for South Asia). This could be due to a relatively localized hydrological cycle or seasonality in the direction of moisture flow. Indeed, also other regions that are not identified as hubs from a degree perspective have relatively many ZLs, namely monsoon (e.g., West Africa and North America) and mountainous regions (e.g., the Alps and the Caucasus),

which are characterized by seasonal, high-intensity moisture flows from ocean-to-land. These could relatively easily generate strong reciprocal dependencies at annual time scales. Therefore, we complemented our yearly results with a finer monthly resolution for the feed-forward loop minus zero loop results (cf. Fig. 2a with Figs. S17–S20). The variations throughout the seasons are significant and are likely due to shifts in the Intertropical Convergence Zone (ITCZ), but potentially also due to the respective monsoon systems in the four focus regions. While the role of the focus regions can strongly vary throughout the year (e.g., AB is a washing machine in November, but a clear lens in most other months, see Fig. S17), the average of the monthly observations matches well with the annually averaged results (see Fig. 2a). This means, also on a monthly resolution, that IA and CR remain washing machines, SA is a weak washing machine, and AB is a lens.

## Discussion

Our network approach reveals features of the global hydrological cycle that have been overlooked in previous analyses. Considering that particularly FFLs have been shown to increase the possibility of cascading tipping in networks of tipping elements[19], we here present an additional argument for assessing the robustness of Earth's major moisture recycling hubs. It is feared that parts of the Amazon rainforest may tip as a result of deforestation and climate change[24,41,42]. The weakening of links in the network may heavily affect moisture flows down the network. Such cascading effects may additionally be amplified by climate change: the propagation of droughts through a network dominated by FFLs expands the area hit by such events and could possibly increase the likelihood of exceeding thresholds for tipping cascades[19]. Previous work found that parts of networks with many FFLs are more likely to spread perturbations and lead to cascading transitions[19]. To examine and compare the spread of such perturbations globally among the (tropical) ecosystems, the notion of motifs could be linked to dynamical-systems studies that include simplified vegetation dynamics of ecosystems[41,43]. Furthermore, as the atmosphere warms, it can contain more water vapor. On a global scale, this increase is predicted to be larger than the concurrent increases in evaporation and precipitation, implying that atmospheric moisture recycling will tend to occur over larger distances[44], altering the topology of Earth's moisture recycling network and the hydrological cycle in general[45].

In contrast to the overwhelming majority of studies employing climate networks, in which the connections in the network are based upon a statistical similarity of measured time series[14,46–50], our network represents directed causal connections. This, first, allows for a straightforward interpretation of the measured results and, second, opens up many pathways for subsequent research: analyzing higher-order network measures, such as betweenness[23,47,51], could identify connectors in the network; investigating the temporal evolution of the moisture recycling network, such as the impact of different phases of the El Niño Southern Oscillation[52], could help explain climatic anomalies; and developing networks using climate-change projections could offer a new view on Earth system functioning under global warming.

## Methods

### UTrack atmospheric moisture tracking model

The UTrack atmospheric moisture tracking model is a novel Lagrangian model that tracks parcels of moisture forward in three-dimensional space[9]. UTrack is the first moisture tracking model to employ ERA5 reanalysis data[8]. The basic principle of the model is that for each mm of evaporation, a certain number of "moisture parcels" is released and subsequently tracked through time and space. At each time step, the moisture budget of the parcels is updated based on evaporation and precipitation at the respective time and location,

meaning that for each location of evaporation, a detailed image of the "footprint" of evaporation can be created. All types of evapotranspiration are included, and here is simply called evaporation.

For each mm of evaporation, 100 parcels are released 50 hPa above the surface height at random spatial locations within each 0.25° grid cell of input evaporation data. The trajectories of the parcels are based on interpolated three-dimensional ERA5 wind speed and wind direction data, which also have a horizontal resolution of 0.25° and consist of 25 pressure layers in the atmospheric column. The spatial coordinates of each parcel are updated at each time step of 0.1 h. Also, at each time step, there is a certain probability that a parcel is redistributed randomly along the atmospheric column such that, on average, every parcel is redistributed every 24 h (see methods section Moisture recycling dataset: validation and uncertainties below for further details). The relative probability of the new position in the atmospheric column is scaled with the vertical moisture profile. Parcels are tracked for 30 days or until 99% of their moisture has precipitated.

To allocate a certain fraction of any moisture parcel to precipitation events at the current time and location, ERA5 hourly total precipitation ($P$) and total precipitable water (TPW) are interpolated to the simulation time step of 0.1 h. The amount of moisture that precipitates at a certain time step equals the amount of precipitation at that time step over the total precipitable water in the atmospheric water column ($P$/TPW). Specifically, precipitation $A$ in mm per time step at location $x, y$ at time $t$ that originated as evaporation from a particular source is described as:

$$A_{x,y,t} = P_{x,y,t} \frac{W_{\text{parcel,t}} E_{\text{source,t}}}{\text{TPW}_{x,y,t}} \qquad (1)$$

with $P$ being precipitation in mm at time step $t$, $W_{\text{parcel,t}}$ (mm) the amount of moisture in the parcel of interest, $E_{\text{source,t}}$ the fraction of moisture present in the parcel at time $t$ that has evaporated from the source, and $TPW$ (mm) the precipitable water in the atmospheric water column. The moisture content of parcels is updated each time step using evaporation and precipitation at its current location:

$$W_{\text{parcel,t}} = W_{\text{parcel,t}-1} + (E_{x,y,t} - P_{x,y,t}) \frac{W_{\text{parcel,t}-1}}{\text{TPW}_{x,y,t}} \qquad (2)$$

The moisture (fraction) that has evaporated from the source is updated as follows:

$$E_{\text{source,t}} = \frac{E_{\text{source,t}-1} W_{\text{parcel,t}-1} A_{x,y,t}}{W_{\text{parcel,t}}} \qquad (3)$$

The moisture flow $m_{ij}$ from evaporation in cell $i$ to precipitation in cell $j$ is aggregated on a monthly basis (mm/month), where $[x, y] \in j$ becomes:

$$m_{ij} = \sum_{t=0}^{\text{month}} A_{j,t} \frac{E_{i,t}}{W_{i,t}} \qquad (4)$$

with $W_{i,t}$ being the tracked amount of moisture from the source cell $i$ at time $t$. These simulations were performed for all evaporation on Earth during 2008–2017. The results were then aggregated on a mean-monthly basis to produce monthly means, and stored at 0.5 degree resolution. This dataset can be downloaded from ref. 53. For details on how to process the data, we refer to the accompanying paper by ref. 3.

## Moisture recycling dataset: validation and uncertainties
As with all moisture recycling simulations, the ones used in this study rely on a number of assumptions that may affect the moisture recycling rates. All offline moisture recycling models use atmospheric model output to simulate the path of evaporation through the atmosphere to the location where it precipitates. Therefore, there are two sources of uncertainty that affect the moisture recycling estimates: (1) the quality of the atmospheric forcing data and (2) the assumptions in the moisture tracking model.

Tuinenburg and Staal (2020)[9] explored these sources of uncertainty for a number of locations globally. The effects of a decrease in the quality of the atmospheric forcing data were most important in the vertical resolution of the atmospheric data: the forcing data should have enough vertical levels to resolve any vertical shear in atmospheric moisture transport. If the forcing data has a low vertical resolution, the moisture tracking model is forced with the mean atmospheric flow over a number of layers. In many regions, there are surface moisture flows that are in a different direction than the moisture flow aloft, resulting in a very small vertically integrated transport, which would distort the simulation of atmospheric moisture transport. Compared to the vertical resolution of the forcing data, the horizontal and temporal resolutions were less important in order to keep errors as small as possible. Because of the importance of this high vertical resolution, it was recommended[9] to use the ERA5 dataset[8] as its forcing dataset, as this currently is the atmospheric reanalysis dataset with the highest vertical resolution.

In addition, the change of ERA-interim to ERA5 resulted in a much better land-surface scheme with monthly varying vegetation and better bare soil evaporation. Also, many more observations are assimilated, which results in a better precipitation product compared to ERA-interim. Following this, the tracking of atmospheric moisture using ERA5 allows for a better quality of the atmospheric moisture cycle than before. But, of course, also the already high horizontal resolution of $0.5° \times 0.5°$ has the limitation that very localized moisture recycling features like orography and locally varying land use cannot be resolved. Out of these reasons, the uncertainty in the evaporation estimates is a lot larger than that in the precipitation estimates, because of the lack of global evaporation measurements and the difficulty in measuring evaporation in general[54,55].

There are also uncertainties due to the assumptions in the moisture tracking model that can be split into a category of simulation assumptions and physical assumptions. The simulation assumptions include model formulation (Eulerian vs. Lagrangian model set-ups), time step lengths, number of parcels released, and types of interpolation. Of these simulation assumptions, the most important aspects were the model formulation, with Lagrangian models better able to resolve complex terrain and atmospheric flows. For the other model assumptions (see methods section UTrack atmospheric moisture tracking model), it was chosen to simulate with the highest level of precision before any more information (e.g., more parcels) would no longer affect evaporation footprints and moisture recycling statistics (see ref. 9 for further details). Even though the ERA5 dataset is known to have some precipitation biases in the tropics, the results of UTrack (forced by ERA5) have recently been validated across the tropics by independent measurements of deuterium excess, a measure of a stable isotope that depends on terrestrial precipitation recycling[56]. UTrack estimates and isotope-based estimates of terrestrial moisture recycling corresponded, especially in tropical rainforests (Kendall's $\bar{\tau} = 0.52$[56]), which are found to be moisture recycling hubs on a global scale.

## Network construction
Motivated by the network-like structure of the data, we here employ a network perspective to study moisture flows. Hence, nodes in such a network are grid cells on a regular spherical grid and edges represent the moisture transported. However, interpreting the dataset directly as a weighted network is neither computationally feasible nor does a weighted network allow for identifying motifs, the building blocks of complex networks[17]. We, therefore, aim for an approach utilizing an unweighted network.

As shown in Fig. S1, moisture recycling strengths are heterogeneously distributed over multiple powers of magnitude. Thus, it is not appropriate to just withdraw the moisture transport volume and include all moisture transport connections within the dataset as equal and unweighted links. Instead, we attempt to highlight the strongest moisture pathways and, thus, the backbone of the Earth's moisture recycling network. To, on the one hand, include as much moisture volume as possible but also keep the absolute volume of moisture transport represented per edge as similar as possible, we decided to include edges in a data-adaptive way: we step-wise include links starting from the strongest and stop this procedure as the total moisture transport volume exceeds the variable threshold $\rho$. The resulting edges then represent the backbone of the global moisture recycling network. In the main text, we have shown the results for a network where all edges together represent $\rho = 25\%$ of the total moisture transport. Here and in the SI figures, we add a sensitivity analysis for $\rho = 20\%$ and $\rho = 30\%$ and find that the results are stable for this broader range of total moisture volume thresholds.

## Network measures and motifs

The topology of an unweighted network is typically encoded in an adjacency matrix **A** with elements $a_{ij}$ indicating if there exists an edge from node $i$ to node $j$ ($a_{ij} = 1$) or not ($a_{ij} = 0$). The degree $k$ of a node $i$ describes the number of adjacent edges pointing towards or away from node $i$. Hence, the in-degree is defined by[25]

$$k_{in}^i = \sum_{i=1}^{N} a_{ji} \qquad (5)$$

and out-degree is defined by[25]

$$k_{out}^i = \sum_{i=1}^{N} a_{ij}. \qquad (6)$$

To further analyze the topology of a network and, in particular, the local connectivity patterns, we study the presence of three motifs—the feed-forward loop, the neighboring loop, and the zero loop.

The feed-forward loop (FFL) consists of three nodes, A, B, and C, where nodes A and C are directly connected via a detour over node B (intermediary node). Therefore, we have two different pathways that focus on node C. Hence, this motif can be referred to as a directed lens, due to its focused flow from two nodes on one singular and its purely directed linkage. This network motif has been studied in the context of tipping elements and has been proven to facilitate tipping cascades by lowering critical thresholds[19]. The zero loop (ZL) is made up of a bidirectional connection of two nodes. In contrast to the FFL, where node A does not receive feedback from node C, here, both nodes are dependent on each other without a preferred direction of network flow. This facilitates tipping to a much lesser degree than the FFL motif[19]. The neighboring loop (NBr) is an extension of the ZL. In this case, there is an additional node connected to one of the nodes of a zero loop. Hence, there is a two-step directionality in the motif, but in contrast to the FFL, this motif is characterized by reciprocity.

We count the number of motifs a certain node is involved in the network. The number of FFLs is counted as the number that a certain node is a so-called "target" node. The target node is the node, on which the triangular structure of the motif is converging to, i.e., the node that has been referred to as node C above. The ZL is a symmetric motif for the two involved nodes. Therefore, the number of ZLs of a certain node in the network is counted directly as the number of bidirectional interactions of the inspected node. Lastly, the number of NBrs of a certain node is the number of being in the center of a neighboring loop. With this procedure, each node is characterized by its number of FFLs, ZLs, and NBrs (cf. ref. [19]).

## Motif strength and their spatially aggregated difference

To assess the presence of motifs and, in particular, their relative frequency, we first determine the numbers of FFLs, ZLs, and NBrs per node. Subsequently, we normalize these counts by the respective maximum to obtain the motif strength, which is shown for each network motif in Fig. S5. In Fig. S5a–c, we display the motifs for the global network, and in Fig. S5d–f for the land-to-land network.

To specifically characterize the focus regions by means of the network topology, we evaluate which motifs dominate in which region. Consequently, we compute the difference of the motif strengths shown in Fig. S5 and reveal the patterns shown in Fig. 2. For spatially aggregated motif strength differences (Fig. 2c, d), we then compute the average of the respective values inside the highlighted boxes.

## Sensitivity to link threshold $\rho$

The network analysis featured in the main text uses those moisture recycling edges that together represent $\rho = 25\%$ of all atmospheric moisture recycling on Earth. As we aimed to focus on the strongest moisture flows, we chose a threshold of $\rho = 25\%$ aggregating the strongest moisture transport pathways. This allows us to reveal the regions of strongest moisture connections, which are located in and close to the tropics, as we expected. Overall, the aim of this thresholding procedure is to utilize a network approach with unweighted edges but also take into account the large spread of moisture recycling strengths. To test the robustness of the results to the threshold value, we here show the same figures as above in the main text but with different thresholds $\rho$. Note that the error bars in Fig. 2 are based on the analysis featured in this part (the resulting differences using thresholds of $\rho = 20\%$ and $\rho = 30\%$).

Figures S6 and S7 show the in- and out-degree of the all-to-all and land-to-land network using a threshold of $\rho = 20\%$ (Fig. S6) and $\rho = 30\%$ (Fig. S7). Note that the color bar has been adjusted as the number of links differs substantially between the networks. The main difference between Figs. S6 and S7 is the greater emphasis on moisture recycling in the mid-latitudes in Fig. S7. This is a direct consequence of considering more, and thus also some weaker, links. Acknowledging this difference, we stress that especially the land-to-land patterns (Figs. S6c, d, S7c, d) are consistent. In particular, the four focus regions, as defined in the main text, stand out as the main global land-to-land moisture recycling hubs. To support this visual analysis of the in- and out-degree pattern, we furthermore compute the motif strengths for both network configurations for quantitative validation of the results.

In line with the main text, we compare the FFL and ZL strength (see Fig. 2a–d). Not only the spatial patterns in our sensitivity analysis agree remarkably well with the results in the main text above, but also the focus regions remain basically the same (cf. Fig. S8 for $\rho = 20\%$ and Fig. S9 for $\rho = 30\%$ with Fig. 2). The only slight change is the shift towards a directed lens (spatially aggregated FFL and ZL strength difference) for the Amazon basin in the all-to-all network for increasing $\rho$ (cf. Fig. S8c vs Fig. S9c vs Fig. 2c). We attribute the overproportional increase of the number of FFLs to those that include at least one oceanic grid cell to this noticeable shift. This underscores our characterization of the Amazon basin as a directed lens.

The spatially aggregated FFL and NBr difference (Figs. S10, S11) is structurally the same as above, where we computed the FFL and ZL difference (see Figs. S8, S9). The spatial patterns and the aggregated values are robust against shifts of $\rho$. However, for the Amazon basin (AB), the number of FFLs increases overproportionally in the all-to-all network when we include more links in our analysis. In other words, the spatially aggregated FFL-strength for AB increases for higher thresholds $\rho$ (cf. Figs. S10c, S11c and Fig. 2g).

## Sensitivity to the size of the focus regions

Another aspect affecting the results is the spatial extent chosen as a focus region (i.e., the rectangles in Fig. 2). Varying the size of these

rectangles affects the spatially aggregated measures. For all focus regions besides the Amazon Basin (AB), the values are not significantly affected by changing the rectangle size, as the values close to the focus regions are either coherently negative, as for the Congo Rainforest (CR) and the Indonesian Archipelago (IA), or close to zero (South Asia: SA). The AB is characterized by positive values (tendency to lensing), whereas the more southern parts along the Andes are marked by more negative (corridor/washing machine) values.

Hence, we assess the stability of the results by using the spatial region covered by the Amazon rainforest (the extent of the Amazon rainforest is based on ref. 6) and compare them to the ones obtained by using the rectangle. The results featured in Fig. S12 indicate that only considering the rainforest-covered parts of the AB leads to similar or even more positive (lensing) values, confirming our conclusions that the Amazon rainforest region functions differently from the other focus regions.

### Notes on maps
This paper makes use of perceptually uniform color maps developed by ref. 57. The underlying world maps have been created by cartopy[58].

## Data availability
The data basis comes from the UTrack hydrological tracking model[9], forced with ERA5 global reanalysis data[8]. The resulting moisture recycling data were available under the https://doi.org/10.1594/PANGAEA.912710, published together with a model description paper by ref. 3.

## Code availability
The code leading to the network construction and motif occurrences requires to be run on a high-performance computer system and has large storage and high RAM requirements. The code and respective support is available from N.W. upon request.

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

## Acknowledgements
N.W. acknowledges support from the European Research Council Advanced Grant project ERA (Earth Resilience in the Anthropocene, ERC-2016-ADG-743080). F.W. is grateful for financial support by the BMBF grant climXtreme (No. 01LP1902J) "Spatial synchronization patterns of heavy precipitation events". A.S. acknowledges support from the Dutch Research Council (NWO) Talent Program Grant VI.Veni.202.170. The authors gratefully acknowledge the European Regional Development Fund (ERDF), the German Federal Ministry of Education and Research, and the Land Brandenburg for supporting this project by providing resources on the high-performance computer system at the Potsdam Institute for Climate Impact Research.

## Author contributions
N.W. and F.W. contributed equally to this study and conducted the network analysis and the simulation runs. N.W. conceived the study and designed the figures with input from all authors. All authors have designed the study. F.W. led the writing of the manuscript with input from all authors. A.S. and O.A.T. developed the atmospheric moisture recycling dataset. A.S. supervised this study.

## Funding

## Competing interests
The authors declare no competing interests.
