## [Peer Review File · Nature Communications]

Network motifs shape distinct functioning of Earth's moisture recycling hubsREVIEWER COMMENTS

Reviewer #1 (Remarks to the Author):

The authors study the spatial patterns in global moisture recycling networks, i.e. moisture flow from evaporation to precipitation between each pairs of grid cells on Earth or on the continents. They provide novel insights by focusing on three particular motifs that they call “Feed-forward loop”, “zero loop” and “neighboring loop”, which are indicators of particular effects that they call “lens”, “washing machine” and “directed corridors”, respectively. By comparing the relative strength of the three motifs, they show specific regions that are characterized by a dominant pattern. In particular, they show that the Amazon basin is characterized by the lensing effect, whereas the other tropical regions are characterized by washing machines or directed corridors.

They interpret the motifs in terms of predictability of moisture recycling changes to perturbations, such as forest loss or climate change. For example, the authors explain that the lensing effect is associated with moisture transport in a certain direction and therefore changes could propagate and lead to tipping points, whereas that the washing machine or directed corridor reflect reciprocal dependencies (or local feedbacks) that can lead to unpredictable effects and/or strengthen hysteresis. While this interpretation is justified by earlier work on the stability and dynamics of complex networks in general, it has not been formally tested with the moisture recycling network presented in this manuscript. Therefore, this aspect appears slightly speculative. It would be interesting to test how perturbations actually propagate using scenarios of deforestation or climate change, although this might be beyond the scope of the presented manuscript. It hence opens novel perspective for further research.

The authors also link their results to climate processes and atmospheric circulation patterns, for example with the reference to the Hadley cycles or the biotic pump hypothesis. It is however unclear whether the spatial and time scale of these climatic processes fit with the study. I would expect that the networks mainly reflect local interactions between neighboring grid cells that contain most of the moisture transport. The spatial scale is mentioned in several places throughout the manuscript, but perhaps it could be further developed.

The presented results complement previous studies. For example, the major sink and source regions of continental moisture recycling have been revealed in earlier studies (e.g. Van de Ent et al. 2010 – see ref 2). Here, the authors use a different methodological approach by calculating in- and out-degree of the land-to-land network (number of total incoming and outgoing links between continental grid cells) to show regions that they refer to as “super-receivers” and “super-distributors”. This provide the “hubs” regions in which the motifs investigated in greater details. The concepts are slightly different and therefore it makes sense to introduce a novel terminology, but the authors could explain better how it relates to earlier concepts. It would make sense also to discuss to what extend the in- and out-degrees for the overall network mainly reflect the spatial patterns in global precipitation and evaporation data, as it is expected that “super-receivers” and “super-distributors” are regions of high precipitation and evaporation, respectively.

While some motifs and measures in moisture recycling networks have already been studied at continental scale (e.g. Zemp et al. 2014 – ref 4), here the authors provide a study at global scale and at higher spatial resolution, which are significant advancements. Compared to previous studies, the networks here are here unweighted, and therefore the links do not reflect the amount of moisture transported. To tackle this limitation, the authors use thresholds to distinguish “strong links” that carry lot of moisture. They provide sensitivity analysis for several threshold values (15 to 35%) for the degree distribution and the spatial patterns (20, 25 and 30%). It is however not entirely clear why the authors decided to test these specific thresholds, and whether the results are expected to change beyond the test interval. The significance of the results is clearly shown by comparing the observed patterns to random networks, which is a strength of the study, although it is not clear for which threshold this has been tested.

The methodology is clear, but should contain further information on how the strength of the motifs was

calculated. More specifically, how was the number of motifs counted? (L. 2080) Do the nodes involved in the motif occupy a specific position, or can they occupy any positions in the motif? I think this deserves more explanation.

Overall, I consider the manuscript to be well written and the results clearly presented.

Reviewer #2 (Remarks to the Author):

Title: Network motifs shape distinct functioning of Earth's moisture recycling hubs

Authors: Nico Wunderling, Frederik Wol, Obbe A. Tuinenburg, Arie Staal

General Comment:

This manuscript provides an analysis of atmospheric moisture at global scale using a network motif approach. The paper is well written and covers a relevant topic. However, I think it is necessary to address some aspects before publication.

Specific Comments:

1. Line 19: Gimeno et al. (2012) also support this:

<https://agupubs.onlinelibrary.wiley.com/doi/10.1029/2012RG000389>

2. Line 49: It should be "atmospheric moisture" instead of "moisture"

3. Line 52: If these figures are cited in this point, they should be Figs. S2 and S3

4. Line 59: There are other studies that reveal the effect of deforestation in land surface moisture fluxes. For instance,

Alves et al. (2017): <https://www.scirp.org/journal/paperinformation.aspx?paperid=74585>

Sierra et al. (2021): <https://link.springer.com/article/10.1007/s00382-021-06025-y>

5. Line 94: It should be the "Hadley Circulation" or the "Hadley Cell".

6. Line 100: It should be "land-to-land connections"

7. Line 102: It should be "land-to-land directedness"

8. Figure 2: The called "Amazon basin" is not only the Amazon. It includes the Orinoco basin and the Guiana's Shield.

9. In Figure 1 Caption correct "all-to-all network", "land-to-land network", "ocean-to-land connectivity", as well as in line 107.

10. Line 111: Molina et al. (2019) and Ruiz-Vasquez et al. (2020) also support this:

<https://agupubs.onlinelibrary.wiley.com/doi/full/10.1029/2018JD029534>

<https://link.springer.com/article/10.1007/s00382-020-05223-4>

11. Lines 112-113: References should be cited here. There are many South American groups with studies in these topics. Please acknowledge regional research, not only European-based studies.

12. Line 116: Builes-Jaramillo et al. (2018) provide evidence for such mechanism:

<https://link.springer.com/article/10.1007/s00382-017-3785-8>

Also, the authors should revise the work by Dominguez et al. (2022). They used WRF-Tracers to study the atmospheric moisture transport in the Amazon from annual to diurnal scales. This is the first study focused on the diurnal cycle of water vapor transport in the Amazon. They identify a dominant role of ET on diurnal variations of atmospheric moisture transport in the region.

<https://agupubs.onlinelibrary.wiley.com/doi/abs/10.1029/2021JD035259>

13. Lines 117-118: Sierra et al. (2021) and Ruiz-Vasquez et al. (2020) provide particular studies of changes in moisture transport under deforestation scenarios in the Amazon.

14. Line 131: The authors should show more clearly how the Himalaya is related to a hotspot of ZLs in South Asia.

15. Line 135: South America has also a monsoon regime that takes place mostly over the Amazon. How could the authors explain the difference with other monsoon regions with ZL conditions?

16. Lines 149-150: Revise Allan et al. (2020) who analyzed the effects of climate change on the water cycle, including atmospheric moisture:

<https://nyaspubs.onlinelibrary.wiley.com/doi/10.1111/nyas.14337>

Reviewer #3 (Remarks to the Author):

Review of NCOMMS-22-10698

The authors analyze the functioning of the global moisture recycling hubs, showing consistency with areas identified previously using other approaches. Different from other methods, in this case the authors present an analysis focused on the interactions. The Land-to-land moisture flow is relevant in the sense that it is directly sensitive to human activities, hence, the study provides a better overview of the relevance the identified hubs have for global moisture distribution. It is understood that there is a limit in the length of the manuscript and that providing details is often limited by the length factor. However, the concepts of connectivity as well as the interpretation for FFLs, NBr and ZLs should be clarified. The regions certainly lie in the convergence zones and the rainfall regime has a direct implication for the locations of the recycling hubs. Given that there are continental scale hubs (AB, CR and IA) while SA has a different nature (continent-water mass wise) it could be relevant to discuss on whether the water-land distribution is important in the analysis. I would recommend to revise the interpretation considering how the monsoon dynamics is related to the hubs, as that may be relevant for the interpretation of the results.

- Line 21: evapotranspiration flow is more appropriate considering that the changes to which the text refer imply water fluxes from the vegetation in the form of transpiration.
- Line 26: some information on the constrains in the estimation of the water fluxes from the surface would clarify the need to improve spatial resolution but also the limitations even relatively high resolution at 0.5 degree may have for some regions.
- Line 36: is this a specific reference to identification of transport processes from the *motifs*?
- Line 58: How to interpret the land-to-land moisture flow for areas that are subject to intense land use change but that are small and feature patched coverage.
- Line 66: is that lensing effect mentioned related to the overall effect of water vapor that may stay circulating (or somehow stagnated) over a specific area due to the mean (air) flow conditions?

- Line 77: Could you clarify the interpretation and differences between the corridors, lenses and washing machines? It seems that differences could be related with different time scales of the moisture supply but is not clear if that interpretation is correct.
- Line 183: 30 days exceed by far residence time of water vapor, meaning that within those 30 days the original moisture input from the air mass is lost and remaining moisture is related to additional inputs. How is the input from (long-range) moisture transport separated from that linked to local sourcing of moisture (e.g evapotranspiration fluxes)?
- Line 186: How is the bias in ERA5 precipitation considered? For tropical areas the bias is seasonally significant compared to in-situ observations, mainly at the hourly scale.
- Line 202: given the simulated period is 2008-2017 the result from that is not exactly a climatology, I suggest using monthly mean instead of monthly climatologies.
- Line 221: it is not clear what you mean by saying that the horizontal and temporal resolutions were less important compared to vertical resolution of the forcing data. Does it imply that surface conditions (e.g vegetation, orography) play a minor role in the forcing? I find this a bit odd. Like the case for the temporal resolution, in some areas recycling is a very local feature that depends on heating and occurs in the scale of hours to few days. The feedback between precipitation peaks and water vapor fluxes (e.g ET) is also sensitive to time evolution of both components (plus the response of soil water content) locally.
- Line 328: Apart from the sensitivity to the size of the regions, is it possible to provide an estimate of the sensitivity to vegetation coverage?

Other questions:

- The hubs respond to the regions of low-level convergence, it may be relevant to explore how the convergence affects the FFLs, NBr and ZLs. That is if they are affected in similar way or not.

- Is there a signal of interannual variability modes in the moisture recycling hubs that could allow an interpretation on how conditions such as ENSO could affect the moisture recycling and how this affects rainfall patterns?
- Thinking of differences in the temporal scale of recycling in terms of weather events and intraseasonal variations, it may be convenient, if possible, to mention if the scale influences the result. In the method it was mentioned that the sensitivity to time scale was low but that would imply that recycling is dominated by seasonality and in some areas the estimation of water vapor fluxes shows a strong response to heavy rainfall events.
- Could the monsoon be a reason for which the IA has a larger fraction as washing machines? I would interpret the monsoon circulation as a circular moisture flow, perhaps that could be considered for the analysis. That is similar for CR when you think of the West African Monsoon. That would imply that recycling could be controlled by the seasonality of the monsoon dynamics while the case of AB is different.

Dear Reviewers,

We are very grateful for the substantial feedback and comments by the reviewers. The suggestions, references and advice have helped to revise and improve our manuscript.

Following the comments of the reviewers, the most significant changes in the revised manuscript version are:

1. **Motifs:** We have clarified the notion of the motifs by dedicating a small section to it, and linking it to past literature.
2. **References:** With the help of the reviewers' comments, we have significantly extended our reference list and found additional evidence for the special role of the Amazon rainforest in comparison to Asian and African tropical forests. We especially expanded our reference list by several sources from local groups.
3. **Time-scale:** Following the reviewers' request to comment on the role of motifs on a seasonal time-scale, we have substantiated our motif analysis by simulating and resolving the four focus regions on a monthly scale. We find that the monthly results vary strongly with respect to the respective season, but the average of the monthly observations matches well with the average results we present in Fig. 2a in the main manuscript. This shows robustness of the yearly results towards finer temporal resolutions. We discuss the monthly results in the manuscript and add the respective additional figures to the supplementary information.

We have also considered all minor points of the reviewers, and have marked the changes in blue in our new version of the manuscript and supplementary information.

We are grateful for this opportunity to improve our manuscript and are confident that our revised manuscript meets the high standards of Nature Communications. We are looking forward to your further feedback.

Sincerely yours,

Nico Wunderling, Frederik Wolf, Obbe A. Tuinenburg, Arie Staal

Reviewer #1 (Remarks to the Author):

The authors study the spatial patterns in global moisture recycling networks, i.e. moisture flow from evaporation to precipitation between each pairs of grid cells on Earth or on the continents. They provide novel insights by focusing on three particular motifs that they call “Feed-forward loop”, “zero loop” and “neighboring loop”, which are indicators of particular effects that they call “lens”, “washing machine” and “directed corridors”, respectively. By comparing the relative strength of the three motifs, they show specific regions that are characterized by a dominant pattern. In particular, they show that the Amazon basin is characterized by the lensing effect, whereas the other tropical regions are characterized by washing machines or directed corridors.

They interpret the motifs in terms of predictability of moisture recycling changes to perturbations, such as forest loss or climate change. For example, the authors explain that the lensing effect is associated with moisture transport in a certain direction and therefore changes could propagate and lead to tipping points, whereas that the washing machine or directed corridor reflect reciprocal dependencies (or local feedbacks) that can lead to unpredictable effects and/or strengthen hysteresis. While this interpretation is justified by earlier work on the stability and dynamics of complex networks in general, it has not been formally tested with the moisture recycling network presented in this manuscript. Therefore, this aspect appears slightly speculative. It would be interesting to test how perturbations actually propagate using scenarios of deforestation or climate change, although this might be beyond the scope of the presented manuscript. It hence opens novel perspective for further research.

ANSWER #1: We agree with this assessment of the reviewer. While earlier studies have indeed indicated a special role (e.g. overexpression in comparison to random networks) for the so-called feed forward loop in real-world networks (Milo et al., 2002, Science; Stouffer et al., 2012, Science), the effect leading to increased vulnerability has been found in random networks and in an application to the Amazon rainforest (Wunderling et al., 2020, Chaos, doi: 10.1063/1.5142827). While this study focused on the role of network motifs in the Amazon rainforest, other Amazon rainforest studies used perturbations arising from climate-change-induced drought effects or deforestation to show the potential threat of perturbations, but did not specifically look into network motifs (Zemp et al., 2017, Nature Communications; Wunderling et al., 2022, PNAS).

Overall, the notion of these two latter studies (Zemp et al., 2017; Wunderling et al., 2022) could be brought together with the earlier study on motifs in the Amazon rainforest, but now on a global (or at least for the four investigated focus areas) rather than on a regional scale for the Amazon rainforest. As the reviewer correctly points out, this has much potential for a follow-up study in this direction, linking climate-change/deforestation-induced perturbations with the notion of network motifs, but would be beyond the goals of this study, we believe. More in general, we hope that the novel approach in our paper will generate new ideas in the community beyond that mentioned above.

We have added a short statement to our manuscript in II 179-183.

The authors also link their results to climate processes and atmospheric circulation patterns, for example with the reference to the Hadley cycles or the biotic pump hypothesis. It is however unclear whether the spatial and time scale of these climatic processes fit with the study. I would expect that the networks mainly reflect local interactions between neighboring grid cells that contain most of the moisture transport. The spatial scale is mentioned in several places throughout the manuscript, but perhaps it could be further developed.

ANSWER #2: We agree with the reviewer that it is worthwhile to elaborate on the large-scale spatial Hadley circulation pattern. We have added the following sentences to our new manuscript version (see II 59-64):

The large-scale tropical moisture recycling patterns are dominated by the Hadley Cell dynamics, with a small convergence zone with intense precipitation. The precipitation that falls in this convergence zone evaporated earlier in a relatively large zone in a band around the convergence zone (Van der Ent et al., 2017, Hess). Due to the large area that contributes evaporation to a small area of intense precipitation, the pattern of out-degree differs systematically from that of in-degree, without marked regions that have outstanding out-degree maxima (see Fig. 1a, c).

The presented results complement previous studies. For example, the major sink and source regions of continental moisture recycling have been revealed in earlier studies (e.g. Van de Ent et al. 2010 – see ref 2). Here, the authors use a different methodological approach by calculating in- and out-degree of the land-to-land network (number of total incoming and outgoing links between continental grid cells) to show regions that they refer to as “super-receivers” and “super-distributors”. This provide the “hubs” regions in which the motifs investigated in greater details. The concepts are slightly different and therefore it makes sense to introduce a novel terminology, but the authors could explain better how it relates to earlier concepts. It would make sense also to discuss to what extend the in- and out-degrees for the overall network mainly reflect the spatial patterns in global precipitation and evaporation data, as it is expected that “super-receivers” and “super-distributors” are regions of high precipitation and evaporation, respectively.

ANSWER #3: The reviewer is correct in outlining the need to compare our research in more depth to related literature, already in the according results section. The data described in Tuinenburg et al. (2020, ESSD) is the basis for this work. Therefore, the identified hubs are indeed expected to reflect the regions of high precipitation and evaporation because they arise purely from our thresholding ($\rho=25\%$) process of actual moisture flow values. However, by turning the moisture flows into a network, there is an entire toolbox of methods, which can be applied to the moisture recycling data, including but not limited to the study of motifs (see e.g. Newman 2018, *Networks*, Oxford University Press; Milo et al., 2002, *Science*).

As a first result, indicating the existence of super-receivers but the absence of super-distributors of moisture confirms the tendency found in Van der Ent et al. (2010, *Water Resource Res.*). This is further confirmed as the reviewer suspected by our ERA5 data: we added a supplementary figure (see **supp. Fig. S4) comparing the ERA5 data on**

land precipitation and land evaporation to our results in Fig. 1b,d. From there, we depart into network measures, especially with respect to motifs, complementing earlier research which focussed on recycling ratios from land/vegetation (e.g. Tuinenburg et al., 2020, ESSD; Keys et al., 2016, PLoS ONE; Van der Ent et al., 2010, Water Resource Res.). Therefore, we expect our results to say more about structural properties of the moisture recycling flows (e.g. through spatial patterns of different motif types) as compared to earlier literature.

We discuss this now in **II 59-64** and **II 73-76**.

While some motifs and measures in moisture recycling networks have already been studied at continental scale (e.g. Zemp et al. 2014 – ref 4), here the authors provide a study at global scale and at higher spatial resolution, which are significant advancements. Compared to previous studies, the networks here are here unweighted, and therefore the links do not reflect the amount of moisture transported. To tackle this limitation, the authors use thresholds to distinguish “strong links” that carry lot of moisture. They provide sensitivity analysis for several threshold values (15 to 35%) for the degree distribution and the spatial patterns (20, 25 and 30%). It is however not entirely clear why the authors decided to test these specific thresholds, and whether the results are expected to change beyond the test interval.

ANSWER #4: We are grateful that the reviewer assesses that our work includes significant advances. Regarding the threshold, we have chosen to include $\rho=25\%$ to identify the main moisture recycling hubs, which are relevant to the entire Earth system. We do this to receive spatial sizes that are similar to the definition of climate tipping elements (see Armstrong McKay et al., 2022, <https://doi.org/10.1002/essoar.10509769.1>). This means, we aim for regions, which are at least of sub-continental scale.

By doing this, we focus on the strongest moisture recycling links only, including all links with a yearly moisture flow above a value of approximately 2 mm/yr (see supp. Fig. S1 at $\rho=25\%$). 2 mm/yr represent on average on a $1^\circ \times 1^\circ$ grid cell $2 \text{ mm/yr} \cdot (50 \text{ km})^2 \approx 5 \cdot 10^9 \text{ liters}$, or in equatorial regions 2 mm/yr represent $2 \text{ mm/yr} \cdot (111 \text{ km})^2 \approx 25 \cdot 10^9 \text{ liters}$ on a $1^\circ \times 1^\circ$ grid cell.

To additionally reflect on the stability of our results, we varied our threshold in the SI between $\rho=20\%$ and $\rho=30\%$ (see supp. Figs. S6-S11). As we expected, the strongest links can be found in and close to the tropics.

Being more strict (i.e. taking values of $\rho < 20\%$), the size of the hubs that we now find is getting smaller up to the point that they disappear, which we do not want. Being more lenient (i.e. taking values of $\rho > 30\%$), we would identify minor hubs besides the major hubs, which we also do not aim for. We have elaborated on this briefly in **I 53-57** and **II 347-350**.

In addition to these heuristic network topology arguments, there are also good reasons to introduce a threshold of 25% when converting the weighted original moisture flow connections into an unweighted network. The very broad distribution of moisture flow values (see supp. Fig. S1) indicates that dropping the actual moisture flow value alone

does not allow for a useful network analysis because then weak moisture flow connections would be counted in the same way as strong moisture flow connections would.

As with all networks and to assure that the links in the unweighted network are meaningful, it is therefore required to introduce a threshold above which connections are counted (and below which connections are set to zero). Since we are interested in analyzing a network and not only singular links, we need to assure that a considerable amount of total moisture flow is represented, while at the same time the number of link densities must allow for a comprehensive network-data analysis.

Combined with our arguments above, we find that the results are valid in the range from $\rho=20\%-30\%$ of the total moisture flow, which means that there are enough links in the network that all represent moisture flows on the right tail of the moisture flow distribution.

The significance of the results is clearly shown by comparing the observed patterns to random networks, which is a strength of the study, although it is not clear for which threshold this has been tested.

ANSWER #5: The threshold for which the random network has been tested against the real-world moisture recycling network is 25%. We thank the reviewer for catching this as this information has indeed been missing from the earlier version of the manuscript, see now II 105-106.

The methodology is clear, but should contain further information on how the strength of the motifs was calculated. More specifically, how was the number of motifs counted? (L. 280) Do the nodes involved in the motif occupy a specific position, or can they occupy any positions in the motif? I think this deserve more explanation.

ANSWER #6: We agree with the reviewer that our methodology of how we count the motifs should be better explained. In the manuscript, we count the number of “target” nodes of the respective motif kind. The target node is the node to which the motif points to (compare Wunderling et al., 2020, Chaos). For the feed forward loop in Fig. 2 for instance, this node is depicted by a forested grid cell. This grid cell is influenced directly by a node (in Fig. 2 shown as a grassland grid cell), but also indirectly via one intermediary node (in Fig. 2 shown as an ocean grid cell). The ZL is a symmetric motif for the two involved nodes. Therefore, the number of ZLs of a certain node in the network is counted directly as the number of bidirectional interactions of the inspected node. Lastly, the number of NBRs of a certain node is the number of being in the *center* of a neighboring loop. With this procedure, each node is characterized by its number of FFLs, ZLs and NBRs.

We have expanded our explanation in the manuscript, see II 325-332.

Overall, I consider the manuscript to be well written and the results clearly presented.

ANSWER #7: We are very grateful for this positive evaluation of our work.

Reviewer #2 (Remarks to the Author):

Title: Network motifs shape distinct functioning of Earth's moisture recycling hubs

Authors: Nico Wunderling, Frederik Wol, Obbe A. Tuinenburg, Arie Staal

General Comment:

This manuscript provides an analysis of atmospheric moisture at global scale using a network motif approach. The paper is well written and covers a relevant topic. However, I think it is necessary to address some aspects before publication.

We are very grateful for this positive assessment of our work, and the comments below to improve our work.

Specific Comments:

1. Line 19: Gimeno et al. (2012) also support this:
<https://agupubs.onlinelibrary.wiley.com/doi/10.1029/2012RG000389>

ANSWER #1: We are thankful for this reference, and are happy to add this suggestion, see I 20.

2. Line 49: It should be "atmospheric moisture" instead of "moisture"

3. Line 52: If these figures are cited in this point, they should be Figs. S2 and S3

ANSWERS #2 and #3: We thank the reviewer for catching this, see I 60 and I 67.

4. Line 59: There are other studies that reveal the effect of deforestation in land surface moisture fluxes. For instance,

Alves et al. (2017): <https://www.scirp.org/journal/paperinformation.aspx?paperid=74585>

Sierra et al. (2021): <https://link.springer.com/article/10.1007/s00382-021-06025-y>

ANSWER #4: We thank the reviewer for these additional suitable references, which we included now in II 73-74.

5. Line 94: It should be the "Hadley Circulation" or the "Hadley Cell".

ANSWER #5: We have adapted this change that the reviewer correctly pointed out (see I 113)

6. Line 100: It should be "land-to-land connections"

7. Line 102: It should be "land-to-land directedness"

ANSWER #6-#7: We agree that this would be a better-readable style of writing. Therefore, we have consistently added hyphens to land-to-land, all-to-all, or ocean-to-land, etc..

8. Figure 2: The called "Amazon basin" is not only the Amazon. It includes the Orinoco basin and the Guiana's Shield.

ANSWER #8: The reviewer is right in pointing this out. Therefore, we have mentioned that we defined the Amazon basin in a broad sense, for instance that it includes the Orinoco basin and Guiana shield (see manuscript II 78-79).

9. In Figure 1 Caption correct "all-to-all network", "land-to-land network", "ocean-to-land connectivity", as well as in line 107.

ANSWER #9: We agree that this would be a better-readable style of writing. Therefore, we have consistently added hyphens to land-to-land, all-to-all, or ocean-to-land, etc..

10. Line 111: Molina et al. (2019) and Ruiz-Vasquez et al. (2020) also support this:

<https://agupubs.onlinelibrary.wiley.com/doi/full/10.1029/2018JD029534>

<https://link.springer.com/article/10.1007/s00382-020-05223-4>

11. Lines 112-113: References should be cited here. There are many South American groups with studies in these topics. Please acknowledge regional research, not only European-based studies.

ANSWERS #10-#11: We agree with this assessment of the reviewer, and are grateful for the additional literature suggestions here and in other places, especially taking into account studies from local groups. Additionally, we added some further references from South American groups replacing European-based studies, as e.g. in II 132-133 or II 136-138.

12. Line 116: Builes-Jaramillo et al. (2018) provide evidence for such mechanism:

<https://link.springer.com/article/10.1007/s00382-017-3785-8>

Also, the authors should revise the work by Dominguez et al. (2022). They used WRF-Tracers to study the atmospheric moisture transport in the Amazon from annual to diurnal scales. This is the first study focused on the diurnal cycle of water vapor transport in the Amazon. They identify a dominant role of ET on diurnal variations of atmospheric moisture transport in the region.

<https://agupubs.onlinelibrary.wiley.com/doi/abs/10.1029/2021JD035259>

ANSWER #12: We thank the reviewer for adding the study from Builes-Jaramillo et al.. Further, the work by Dominguez et al. (2022, J. Geophys. Res. Atm.) is indeed a very interesting study on the timescales of ET effects in the Amazon rainforest. While the study goes down to diurnal timescales revealing very different diurnal patterns for different regions in the Amazon (see their Figs. 5-8), the seasonal to yearly patterns match with what we would expect (see Figs. 2 and 3 in their paper), and has been shown

in an earlier study (see Wunderling et al., 2022, PNAS, supplementary Figs. S1-S3). We have added this reference to our discussion, see II 136-138.

13. Lines 117-118: Sierra et al. (2021) and Ruiz-Vasquez et al. (2020) provide particular studies of changes in moisture transport under deforestation scenarios in the Amazon.

ANSWER #13: We agree and have added these references to I 138 in our new manuscript version.

14. Line 131: The authors should show more clearly how the Himalaya is related to a hotspot of ZLs in South Asia.

ANSWER #14: We extend our analysis and are happy to show a regional plot of the South Asia/Himalaya region (Fig. R1), where we overlaid a topographic height elevation map with our normalized motif map (compare to Fig. 2a,b). It can be seen that especially the eastern, but also the southwestern part of the Himalaya mountains are dominated by ZLs. This suggests that regions close to high elevations where atmospheric moisture transport reaches mountains are regions that are prone to many ZLs in comparison to FFLs. We have added this figure to our supplement and refer to it in the new version of the main manuscript (see II 152-153 and new supp. Fig. S16).

Fig. R1: FFLs minus ZLs for the Himalaya region (South Asia in the manuscript), indicating that the Himalayan mountains are the reason for an increased number of ZLs (land-to-land connections only).

15. Line 135: South America has also a monsoon regime that takes place mostly over the Amazon. How could the authors explain the difference with other monsoon regions with ZL conditions?

ANSWER #15: This is a good point to clarify. Also in Fig. 2a,b, the exact location of the mountainous regions in the Andes is dominated by ZLs. Further, we believe that the strong FFL-dominated region in the Amazon rainforest is due to the main atmospheric moisture recycling mechanism where the main wind directions runs from the Atlantic (east) towards the Andes (west) and is there bent southward, as has, for instance, been described in Dominguez et al. (2022, J. Geophys. Res. Atm.).

Also, it is important to note that Fig. 2 is of relative and not of absolute nature, meaning that the number of FFLs and ZLs are normalized and subtracted from each other. Lastly, the South American monsoon as an atmospheric mechanism is important for this region to show up as one of our four moisture recycling hubs.

The overall cause for this exceptional behavior of the Amazon rainforest moisture recycling hub is not finally known (as we stated in II 132-133), but surely an interesting feature to further investigate. Part of this exceptional behavior might be due to the north-south alignment of the Andes, the ITCZ, or due to the biotic pump hypothesis, diurnal atmospheric changes, or potentially even anthropogenic influence (see e.g. Saatchi et al., 2021, One Earth; Dominguez et al., 2022, J. Geophys. Res. Atm.; Makarieva & Gorshkov, 2007, Hydro. Earth Sys. Sci.).

While the final reason is unknown, we have argued along the lines of the above, see our new manuscript II 124-138.

16. Lines 149-150: Revise Allan et al. (2020) who analyzed the effects of climate change on the water cycle, including atmospheric moisture:

<https://nyaspubs.onlinelibrary.wiley.com/doi/10.1111/nyas.14337>

ANSWER #16: We thank the reviewer for pointing us to this important review article, which we are happy to include (see II 186-187).

Reviewer #3 (Remarks to the Author):

The authors analyze the functioning of the global moisture recycling hubs, showing consistency with areas identified previously using other approaches. Different from other methods, in this case the authors present an analysis focused on the interactions. The Land-to-land moisture flow is relevant in the sense that it is directly sensitive to human activities, hence, the study provides a better overview of the relevance the identified hubs have for global moisture distribution. It is understood that there is a limit in the length of the manuscript and that providing details is often limited by the length factor. However, the concepts of connectivity as well as the interpretation for FFLs, NBr and ZLs should be clarified.

ANSWER #1: We have amply expanded our afore short introduction to motifs in our new version of the manuscript following the advice of the reviewer. More specifically and in detail, we responded to the raised points and elaborated on them in our Answers #5, #7, #8 below, and following a comment of reviewer #1 also in II 325-332.

The regions certainly lie in the convergence zones and the rainfall regime has a direct implication for the locations of the recycling hubs. Given that there are continental scale hubs (AB, CR and IA) while SA has a different nature (continent-water mass wise) it could be relevant to discuss on whether the water-land distribution is important in the analysis. I would recommend to revise the interpretation considering how the monsoon dynamics is related to the hubs, as that may be relevant for the interpretation of the results.

ANSWER #2: We agree and have added a discussion on temporal scales and monsoon dynamics in our Answers #9 and #18 below.

- Line 21: evapotranspiration flow is more appropriate considering that the changes to which the text refer imply water fluxes from the vegetation in the form of transpiration.

ANSWER #3: It is indeed better to change the according sentence, see I 22.

- Line 26: some information on the constrains in the estimation of the water fluxes from the surface would clarify the need to improve spatial resolution but also the limitations even relatively high resolution at 0.5 degree may have for some regions.

ANSWER #4: We agree with the reviewer. The change of ERA-interim to ERA5 resulted in a much better land-surface scheme with monthly varying vegetation and better bare soil evaporation. Furthermore, the many more observations are assimilated which results in a better precipitation product compared to ERA-interim. The newer dataset (e.g. Tuinenburg et al., 2020, ESSD) is using the most up-to-date reanalysis products from ERA5 (Hersbach et al., 2020, Quart. J. Roy. Met. Soc.). With this new product, the atmospheric moisture cycle is better simulated than for previous products that used

pre-ERA5 forcing. But still, even with a resolution of $0.5^{\circ} \times 0.5^{\circ}$ there are still limitations in resolving very localized moisture recycling features like orography that affects spatial precipitation variability and locally varying land use, which could affect the ERA5 evaporation estimates. Out of these reasons, the uncertainty in the evaporation estimates is a lot larger than that in the precipitation estimates, because of the lack of global evaporation measurements and the difficulty in measuring evaporation in general (Mueller et al., 2011, *Geophys. Res. Lett.*; Harding et al., 2014, *Data, Models and Uncertainties in the global water cycle*, Springer Water). We have added this to our new version of the manuscript in **II 259-268**.

Important to note: For the results of our study, revealing the main moisture recycling hubs, it is sufficient to keep the resolution at $1^{\circ} \times 1^{\circ}$. This also saves computational resources, which are already very demanding on a 180×360 grid (and requires using large-RAM nodes on a high performance computer cluster).

- Line 36: is this a specific reference to identification of transport processes from the Motifs?

ANSWER #5: Yes, in earlier research it has been shown that motifs play a special role in transporting information and in transport processes in general. Especially, the feed-forward loop (FFL) motif has been studied extensively in earlier research (among others, see Milo et al., 2002, *Science*; Alon, 2007, *Nat. Rev. Genet.*; Gorochowski et al., 2018, *Sci. Adv.*), for instance finding that this motif is significantly overexpressed in real-world networks as compared to random networks.

This has led the research community to believe that FFLs have a special role in information transport processes in larger complex networks. However, while this transport process can be positive in some contexts (e.g. finding websites through links in the world-wide-web), it might be turned negative in other contexts (e.g. deforestation and subsequent loss of moisture in the overall recycling network). Besides FFLs, also other small scale motifs like zero loops (ZLs) have been found to be important for cascading effects (see e.g. Wunderling et al., 2020, *Chaos*). In this sense, motifs and especially FFLs are tracers of enhanced (gain/loss of) information transport in complex networks. We have expanded this section and added several further references to our manuscript, see **II 40-47**.

- Line 58: How to interpret the land-to-land moisture flow for areas that are subject to intense land use change but that are small and feature patched coverage.

ANSWER #6: If land use change is playing out in areas significantly smaller than our resolution ($1^{\circ} \times 1^{\circ}$), this cannot be resolved. Also, if the land coverage is differently patched on a scale significantly below our employed resolution, those patches cannot individually be resolved, but these sub-resolution-scale effects would be averaged over. Having said this, it is important to note that our model is designed to reveal large-scale effects of global moisture recycling hubs, large-scale deforestation and irrigation

patterns, while it should not be used to investigate very localized effects of atmospheric moisture transport changes (see II 73-74).

- Line 66: is that lensing effect mentioned related to the overall effect of water vapor that may stay circulating (or somehow stagnated) over a specific area due to the mean (air) flow conditions?

ANSWER #7: The reviewer raises an important point to be clarified here. The FFL motifs are not related to circulating moisture that is stagnated in the atmosphere because the FFLs indicate regions where moisture has a preferential direction (lens) of movement instead of a stagnation. For instance, in the Amazon rainforest, this preferential direction of movement reaches South America from the Atlantic and is then moving further westward (Zemp et al., 2014, *Atm Chem Phys*; Dominguez et al., 2022, *J. Geophys. Res. Atm.*; Wunderling et al., 2022, *PNAS*). However, a circulating effect can be expected for ZLs, where grid cell A is connected to grid cell B and vice versa (washing machine). But also in the case of the ZLs we are not necessarily observing stagnated water in the atmosphere because we are investigating an average year consisting of the monthly means from 2008-2017. We clarified this in the main manuscript (see II 29-32). We can therefore not resolve effects below the average residence time (typically on the order of 1-2 weeks) of stagnated moisture in the atmosphere.

Line 77: Could you clarify the interpretation and differences between the corridors, lenses and washing machines? It seems that differences could be related with different time scales of the moisture supply but is not clear if that interpretation is correct.

ANSWER #8: We agree with the reviewer that a clarification of our three notions of what FFLs (lenses), ZLs (washing machines), and NBRs (corridors) mean is key for our work and is therefore worthwhile to expand on. Please see the new version of the manuscript II 82-99 and below here:

1. FFLs involve three nodes linked in a directed triangular way, creating a lensing effect of moisture flow. This lensing effect is caused by a start node that is linked to a target node directly, but also via a one-node-detour through an intermediary node. This means that node A (start node) is linked to node B (target node) directly, but also via the one-node-detour via node C (intermediary node). Through this two-way interaction, moisture flow changes from source to target would be larger than they would appear if only direct flows would be analyzed. Therefore, FFLs are directed lenses in our notion.
2. ZLs consist of two nodes that are connected bidirectionally in a reciprocal manner, i.e. node A is connected to node B and vice versa. In other words, they show a feedback between two nodes in the network, such that hydrological change in one node is transferred to the other, and by the reciprocal connection potentially stabilized at a higher level as compared to the FFL. As the ZLs are not directed motifs, we call them washing machines.

3. NBRs indicate directed reciprocity: they describe a zero loop connected to an additional source node that induces a directed moisture flow towards the ZL. This means a node A (start node) is coupled to a node B (target node), and this node B is part of a zero loop together with node C. Like FFLs, they involve three nodes and have a flavor of directedness in their topology. Thus, they share features of both node-to-node dependency and a lensing effect. NBRs imply that land-cover or atmospheric change in the source node could disrupt the hydrological feedback between two other bidirectionally coupled nodes. Since NBRs are a directed extension of zero loops, we call them directed corridors.

For the notion of the motifs, we were building on an earlier paper of the coauthors (Wunderling et al., 2020, Chaos, especially Fig. 1).

—

Unless the reviewer meant monsoon or ITCZ dynamics within a year (see below), the differences between the motifs cannot be due to different timescales because we evaluate all our data for the same time, i.e. as the aggregation of 12 months (taken from averaging the years 2008-2017, following the prepared data by Tuinenburg et al., 2020, ESSD). Furthermore, while diurnal timescales of moisture recycling flows have been investigated for the Amazon rainforest (Dominguez et al., 2022, J. Geophys. Res. Atm.), rather the seasonal to yearly patterns match with what we would expect (see Figs. 2 and 3 in Dominguez et al., 2022, J. Geophys. Res. Atm.). This has been shown in an earlier study on the Amazon rainforest (Wunderling et al., 2022, PNAS, see their supp. Figs. S1-S3). Please also compare to our **answer #12 to reviewer #2**.

Monsoon dynamics: In our former version of the manuscript, we did not investigate monthly resolutions of motifs and could therefore not reveal monsoon dynamics in our four focus regions. However, we now substantiated our analysis by supplying monthly averages where we investigate the different motifs in the four focus regions. We find considerable differences depending on the season, indicating the shift of the ITCZ over the year, and indicating a potential role of the monsoon system, however, this remains speculative in our opinion. Please see our response **answer #18 below** for a detailed discussion (and the attached Figs. **R3-R6** below), as well as the manuscript **II 158-167** and **supplementary Figs. S17-S20**.

- Line 183: 30 days exceed by far residence time of water vapor, meaning that within those 30 days the original moisture input from the air mass is lost and remaining moisture is related to additional inputs. How is the input from (long-range) moisture transport separated from that linked to local sourcing of moisture (e.g. evapotranspiration fluxes)?

ANSWER #10: The usual residence time for water in the atmosphere is on the order of 8-10 days, and this is reflected in moisture-tracking results such as ours (Van der Ent & Tuinenburg 2017 HESS). We track moisture parcels for up to 30 days or until 99% of its

moisture has rained out, but after 30 days, in most cases more than 99% of its moisture has already been allocated to precipitation (Tuinenburg & Staal 2020 HESS; example see below in Fig. R2). After the start of the simulation (i.e. the tracking of one moisture parcel from one location), no additional input is added to this parcel; it only loses moisture depending on precipitation events at the hour and grid cell that the moisture parcel is then located in. Because the parcels lose moisture progressively (if precipitation occurs) and all locations of precipitation are registered, long-range and short-range moisture transport links are separated automatically.

Fig. R2: Example of moisture tracking as modeled by UTrack for the example of Utrecht in the Netherlands (small red dot in the figure). Upper panel: Origin of the moisture raining down over Utrecht (evaporation footprint); Lower panel: Location, where evaporation from Utrecht lands (precipitation footprint). Figure taken from Tuinenburg et al. (2020, ESSD, their Fig. 1).

- Line 186: How is the bias in ERA5 precipitation considered? For tropical areas the bias is seasonally significant compared to in-situ observations, mainly at the hourly scale.

ANSWER #11: It is known that ERA5 has some biases in tropical areas as the reviewer correctly points out. However, ERA5 precipitation data are the only option to use without introducing internal inconsistencies, as we rely on ERA5 for the atmospheric simulations. Due to the high vertical resolution, it is advisable to use ERA5 over earlier reanalysis products because ERA5 has the highest vertical resolution. Furthermore, the results of the moisture tracking model UTrack, which is forced by ERA5 data, has been validated by deuterium excess measurements, and good agreement is found in the tropics (Cropper et al., 2021, npj Clim. and Atm. Sci.; see our additions in II 269-283). Therefore, the ERA5 precipitation biases do not introduce unphysical moisture recycling values and we believe our choice of taking ERA5 as the base data source is valid. We discuss the uncertainties of the moisture recycling data set specifically related to the quality of the atmospheric moisture forcing data (see II 240-283, and in particular II 247-268).

- Line 202: given the simulated period is 2008-2017 the result from that is not exactly a climatology, I suggest using monthly mean instead of monthly climatologies.

ANSWER #12: Agreed. What we meant with monthly climatologies are monthly means, which we corrected in the manuscript (see I 236).

- Line 221: it is not clear what you mean by saying that the horizontal and temporal resolutions were less important compared to vertical resolution of the forcing data. Does it imply that surface conditions (e.g. vegetation, orography) play a minor role in the forcing? I find this a bit odd. Like the case for the temporal resolution, in some areas recycling is a very local feature that depends on heating and occurs in the scale of hours to few days. The feedback between precipitation peaks and water vapor fluxes (e.g. ET) is also sensitive to time evolution of both components (plus the response of soil water content) locally.

ANSWER #13: We meant that the vertical resolution (of atmospheric pressure layers) is key for our moisture recycling algorithm to produce as small errors as possible. Therefore, deviations from an accurate vertical resolution lead to larger errors than deviations in the horizontal resolution or the temporal resolution does. This is also the main advantage of ERA5 over earlier reanalysis data sets as we outlined in this paragraph.

Of course, at a certain point when horizontal resolution or temporal resolution become too coarse, then also expected errors become large. However, this is not the case with the ERA5 product and the small time steps of the UTrack moisture tracking model (15 minutes).

- Line 328: Apart from the sensitivity to the size of the regions, is it possible to provide an estimate of the sensitivity to vegetation coverage?

ANSWER #14: This is a very interesting research question. Unfortunately this is not possible with the current setup because there is no model yet, which models vegetation dynamically, by at the same time dynamically adapting the moisture recycling network. There are efforts undertaken at the moment, which aim to couple the dynamic vegetation model LPJmL together with the moisture recycling network used in this study. This would allow us to probe different vegetation covers on the moisture transport. While this would already be a large step forward, it still wouldn't allow for a dynamic atmospheric moisture recycling network, which is even one step further.

Apart from that, it would also be possible to account for different types of transpiration (e.g. tree transpiration or interception evaporation) in the underlying moisture recycling dataset, allowing for hypothesis testing of different vegetation covers globally. However, this data are also not yet available on a global scale, and can therefore not be used in our study.

Overall, we agree with the reviewer that it would be very worthwhile to quantify the sensitivity of different vegetation covers on our results and to inspect how our results would change depending on the vegetation cover, but this is beyond the scope of this manuscript.

Other questions:

- The hubs respond to the regions of low-level convergence, it may be relevant to explore how the convergence affects the FFLs, NBr and ZLs. That is if they are affected in similar way or not.

ANSWER #15: We agree that this could be a relevant aspect. In the manuscript, we note the general tropical patterns of network characteristics in and around the convergence zones (see II 59-67). A more in-depth analysis of these characteristics requires studying the network characteristics on a monthly timescale to make sure the seasonal variability of convergence zones is properly assessed. Doing this for every motif in every month is beyond the current manuscript, and will be part of future work.

However, we have additionally checked for potentially important seasonality dynamics (which could be related to monsoon systems) on a monthly time-scale, evaluating whether our four focus regions keep their role as “washing machines” or “lenses” also on a monthly time scale (see our answer #18, and Figs. R3-R6 below).

- Is there a signal of interannual variability modes in the moisture recycling hubs that could allow an interpretation on how conditions such as ENSO could affect the moisture recycling and how this affects rainfall patterns?

ANSWER #16: This cannot be explored with the currently available data in this research because those are the averages from the years 2008-2017. Those years include years with strong El-Niño years (e.g. 2010, 2015+2016) but also years without an El-Niño event. However, we thank the reviewer for this very relevant idea and worthwhile analysis for future work, which we added to the outlook section in our discussion (see II 195-196).

- Thinking of differences in the temporal scale of recycling in terms of weather events and intraseasonal variations, it may be convenient, if possible, to mention if the scale influences the result. In the method it was mentioned that the sensitivity to time scale was low but that would imply that recycling is dominated by seasonality and in some areas the estimation of water vapor fluxes shows a strong response to heavy rainfall events.

ANSWER #17: This is a valid point and seasonality indeed has an influence on the results. In this study we were looking at a decadal mean of the atmospheric moisture recycling flows for the years 2008-2017. As explained above, the sensitivity to the temporal scale is valid for the ERA5 forcing data set in comparison to its vertical resolution (compare to our answer #13).

We agree with the reviewer that it would be worthwhile to extend our analysis to a monthly resolution. Also following the next comment by the reviewer, we therefore substantiated our analysis by the evaluation of motifs in the according average months (i.e. January 2008-2017, February 2008-2017, ...) for all the four focus regions (see **Figs. R3-R6** below). Our results show that the occurrence of motifs strongly differs between the seasons. However, the average of the twelve months matches the average structure well, so a finer temporal resolution reveals more details throughout the seasons, but also confirms our results on a yearly average (see **Fig. 2a** in the main manuscript). Please find a more detailed discussion in our response in **answer #18**.

- Could the monsoon be a reason for which the IA has a larger fraction as washing machines? I would interpret the monsoon circulation as a circular moisture flow, perhaps that could be considered for the analysis. That is similar for CR when you think of the West African Monsoon. That would imply that recycling could be controlled by the seasonality of the monsoon dynamics while the case of AB is different.

ANSWER #18: We agree with the reviewer that an additional analysis on the monthly development of the motifs would be worthwhile to undertake, discussing the seasonal differences with respect to the yearly average. Therefore, we substantiated our analysis by computing and adding a seasonally resolved plot of the feed-forward loop minus zero loop plots for all the four focus regions in their all-to-all variant (see **R3-R6** below).

For the four focus regions, the changes throughout the different months could be due to shifts in the ITCZ during the year. While we cannot prove this in this research, the seasonal shifts indeed also seem to show an imprint of the respective monsoon system in the four focus regions, as the reviewer suspected above.

Further, the role of the four focus regions can strongly vary throughout the season, e.g. the Amazon basin is a washing machine in November, but a clear lens in most other months (see Fig. R3). However, the average of the monthly observations matches well with the average results we present in **Fig. 2a** in the main manuscript. This results shows robustness from the yearly towards the finer monthly temporal resolutions. This means IA and CR are washing machines also on a monthly resolution, while AB is a lens. Lastly, SA is in between these two groups of focus regions, leaning a bit more towards being a washing machine (weak washing machine).

We added a paragraph in the main manuscript **II 158-167**, and added the **supp. Figs. S17-S20** (= Figs. R3-R6 below) to the supplementary information.

Fig. R3: FFL and ZL strength throughout the average months from the years 2008-2017 for the Amazon basin (see rectangle) and the larger South American region (compare to Fig. 2a in the main manuscript).

Fig. R4: FFL and ZL strength throughout the average months from the years 2008-2017 for the Congo Rainforest (compare to Fig. 2a in the main manuscript).

Fig. R5: FFL and ZL strength throughout the average months from the years 2008-2017 for South Asia (compare to Fig. 2a in the main manuscript).

Fig. R5: FFL and ZL strength throughout the average months from the years 2008-2017 for the Indonesian Archipelago (compare to Fig. 2a in the main manuscript).